# Study on the Mechanical Properties of Perforated Steel Plate Reinforced Concrete

**DOI:** 10.3390/ma15196944

**Published:** 2022-10-06

**Authors:** Chunbao Li, Gaojie Li, Liang Zheng, Xiaohui Liu, Shen Li, Xukai Wang, Valentina Y. Soloveva, Hojiboev Dalerjon, Zhiguang Fan, Pengju Qin

**Affiliations:** 1College of Pipeline and Civil Engineering, China University of Petroleum (East China), Qingdao 266580, China; 2China Petroleum LONGWAY Engineering Project Management Co., Ltd., Langfang 065000, China; 3Qingdao Urban Development Group Co., Ltd., Qingdao 266061, China; 4Construction Project Management Branch of China National Petroleum Pipeline Network Group Co., Ltd., Langfang 065001, China; 5Emperor Alexander I St. Petersburg State Transport University, St. Petersburg 190031, Russia; 6Mining-Metallurgical Institute of Tajikistan, Buston City 735730, Tajikistan; 7Henan Huatai New Material Technology Corp., Ltd., Nanyang 473000, China; 8College of Civil Engineering, Taiyuan University of Technology, Taiyuan 030024, China

**Keywords:** perforated steel plate reinforced concrete, compression test, pull-out test, optimal parameters

## Abstract

In this paper, the mechanical properties of perforated steel plate reinforced concrete were studied. Through the compression test of the specimen, the failure mode, the compressive ultimate bearing capacity, and the stress–strain curve of the specimen were obtained. The results show that the compressive strength of perforated steel plate reinforced concrete is twice that of the same grade of plain concrete; through the pull-out test of the specimen, the failure mode and the ultimate uplift bearing capacity were obtained. The finite element software ANSYS was used to simulate the perforated steel plate reinforced concrete specimen, and the results show that the model is reliable. Through the range analysis method, the influence degree of the three factors of the thickness of the perforated steel plate, the hole diameter, and the hole spacing on the compressive strength and the ultimate bearing capacity of the pull-out was studied, and the optimal solution was obtained. The analysis results show that the order of the three factors on the compression and pull-out tests is: the plate thickness of the perforated steel plate > the hole diameter > the hole spacing; the optimal combination of the compressive strength of the perforated steel plate reinforced concrete specimen is that the thickness of the perforated steel plate is 0.75 mm, the diameter of the perforated steel plate is 15 mm, and the spacing of the perforated steel plate is 5 mm; the optimal combination of the ultimate bearing capacity of the pull-out is that the thickness of the steel plate with holes is 1.0 mm, the diameter of the steel plate with holes is 15 mm, and the spacing of the steel plate with holes is 15 mm.

## 1. Introduction

Steel structures are booming in this era, but some problems remain in their engineering application, such as low rigidity and poor fire resistance in high-rise buildings [1]. With the increase in building floors, reinforced concrete structures are prone to form short columns that are unfavorable for anti-seismicity and need to be controlled by some structural measures [2]. A concrete-filled steel tube is a kind of composite structure formed by filling steel tubes with concrete [3]. According to the different cross-sectional forms, they are divided into circular, square, and polygonal concrete-filled steel tubes [4]. The concrete-filled steel tube has a good economic effect. It can save 50% of the steel under the conditions of a similar weight and the same bearing capacity. The weight of the concrete and the components will be reduced by about 50% under the conditions of a similar amount of steel and the same bearing capacity [5,6,7,8,9].

Klppol et al. [10] carried out axial compression tests on 24 concrete-filled circular steel tube columns and derived the calculation formula for the bearing capacity of the concrete-filled circular steel tube columns. Furlong [11] conducted axial compression and eccentric compression tests on 52 concrete-filled steel tube short columns. He observed the damage to the specimens and recorded the corresponding ultimate bearing capacity and put forward the formulas for the axial compression bearing capacity and the ultimate bearing capacity of concrete-filled square steel tube columns. Gardner et al. [12] conducted axial compression tests on 22 concrete-filled circular steel tubes and found that when the concrete-filled steel tube column reached the ultimate bearing capacity, the steel had reached the state while the concrete still had not. Axial compression and compression-bending-shear tests on concrete-filled steel tubes were conducted by Tomii et al., and the influence of the section aspect ratio on the mechanical properties of concrete-filled steel tubes was obtained [13].

Axial compression tests on 57 concrete-filled steel tubular columns were conducted by Cai et al. to study the influence of hoop coefficients on the ultimate bearing capacity of concrete-filled steel tubes, and the calculation formula for the ultimate bearing capacity was proposed [14]. The compression-torsion, bending-torsion, compression-bending-torsion, and compression-bending-shear tests on concrete-filled steel tubes were conducted by Han et al. to study the working mechanism of concrete-filled steel tubes in complex stress states, and the calculation formula of the ultimate bearing capacity under various stress states was deduced [15,16,17,18]. Tan et al. [19,20] conducted axial compression tests on 18 ultra-high-strength circular steel tube columns and found that they did not deform significantly when they reached the ultimate load. The axial compression and eccentric compression tests on 14 concrete-filled steel tube columns with restraint rods were conducted by Cai et al. [21] to investigate the influence of the material strength and restraint rod parameters on the concrete-filled steel tube. The result was that the existence of restraint rods can improve the ductility and ultimate bearing capacity of the components [21,22,23].

Mansour et al. proposed equations that were derived based on the empirical models found in the literature and were set to estimate the shear capacity of normal-strength RC beams without stirrups [24]. Continuous RC beams containing openings were the most affected among the analyzed locations. The reduction in load capacities ranged from 7.3 to 66.1% compared to the solid beam [25]. Sakr et al. investigated the behavior of RC beams strengthened in shear capacity with UHPFRC. The results showed that strengthening RC beams using two UHPFRC plates improved the load-carrying capacity by 145% in comparison to the control beam [26]. Hamoda et al. experimentally investigated the behavior of steel I-beams with/without high-strength bolted connectors embedded in the normal/steel fiber-reinforced concrete (SFRC). Composite action can exist not only in the composite structures but also when casing the critical joints with bolted connectors [27]. A 3D numerical finite element model (FEM) is proposed to simulate the bond behavior between the concrete and the UHPFRC plates, and this study reviews the analytical models that can predict the relationship between the maximum bond stress and the slip for the strengthened concrete elements [28]. The viscous UHPFRC material can fill the pores and the concaves on the NSC surface, causing the shear connectors to enhance the bond performance [29]. The steel connectors prevented the debonding failure pattern and enhanced both the ultimate failure load and the ductility index [30].

In order to promote the building industrialization, this paper introduces a new type of perforated steel plate reinforced concrete composite structure which incorporates the advantages of the concrete-filled steel tube structure. A compression test on 54 specimens and a pull test on 27 specimens were performed. Three factors, including the thickness of the steel plate, the hole diameter, and the hole spacing, and three levels for each factor were designed in the orthogonal test. Finite element analysis and data processing were performed, and the validity of the numerical simulations was verified by the test data. Through range analysis, the influence of these three factors on the compressive strength and the ultimate pull-out bearing capacity was quantified, and the optimum combinations were determined.

## 2. Equipment and Methods

### 2.1. Test Equipment

(1) The compression and pull-out tests were performed on the microcomputer-controlled electro-hydraulic servo universal testing machine which produced by Changchun Sinter Testing Machine Co., Ltd. in Changchun, China.

(2) The pull-out tests simulated the bond slip between the perforated steel plate and the concrete. Special fixtures were designed. The clamp was 20 mm in thickness, the upper hole was 20 mm in diameter, and the width of the lower opening was 40 mm. The pull-out test of the perforated steel plate reinforced concrete is shown in Figure 1.

(3) Prior to concreting, the perforated steel plate was pre-placed at the center line of the specimen by a specially designed mold. The details of mold are shown in Figure 2a. Two 1.674 m-long perforated angle steels were welded together as one unit and five units were finally welded together. After the perforated steel plates were inserted into the corresponding position of the test mold, the perforated steel plates were secured with bolts. The details of the test mold are shown in Figure 2b.

### 2.2. Preparation and Materials

#### 2.2.1. Preparation of Perforated Steel Plate Reinforced Concrete Specimen

In the experiment, three replicate specimens were selected in each mechanical performance test to eliminate the experimental error. In each test, the compressive or pull-out strength values were the average value of three replicate specimens, with an accuracy of 0.1 MPa [31].

If the difference between the maximum and minimum values measured is greater than 15% of the median value, the data measured in this group are not available, and the test should be retested [31].

Ordinary Portland cement produced by China Resources Cement Holding Co., Ltd. in Shenzhen, China [32] with a strength grade of C30 was used to prepare perforated steel plate reinforced concrete specimens with dimensions of 150 mm × 150 mm × 150 mm for the compression test. The specimens are shown in Figure 3a. In the pull-out test, the perforated steel plate was 150 mm × 150 mm × 350 mm, and the concrete specimens were 150 mm × 150 mm × 150 mm. The pull-out specimen is shown in Figure 3b. According to the current research on the mechanical properties of steel–concrete composite members [21,22,23,33], and combined with engineering experience, the perforated steel plates were 0.5 mm, 0.75 mm, and 1.0 mm in thickness. Considering the diameter of the mixture in the concrete, the perforated steel plates were 15 mm, 20 mm, and 25 mm in diameter. According to the staggered arrangement design of the holes and the value range of the diameter, the hole spacings were 5 mm, 10 mm, and 15 mm, respectively.

#### 2.2.2. Materials

The steel produced by Qingdao Xiangxiong Materials Co., Ltd. in Qingdao, China used in the perforated steel plate reinforced concrete specimen designed in this paper is Q235 steel, and the concrete used in the test is made according to its compressive strength of C30.

We took samples from the same batch of steel and concrete and conducted the concrete compression test with a concrete cube specimen according to GB50081-2019 [31]. The measured compressive strength was 29.20 MPa, reaching 97.33% of the standard value of the design grade. According to GB/T 228.1-2021 [34], through the tensile test at room temperature on 0.5 mm-, 0.75 mm-, and 1.0 mm-thick steel plate specimens, the yield strengths were 235.7 MPa, 235.3 MPa, and 235.2 MPa, respectively, and the ultimate strengths were 290.1 MPa, 290.3 MPa, and 291.3 MPa, respectively.

### 2.3. Test Method

#### 2.3.1. Parameter Design

In order to carry out the experiment comprehensively and efficiently, the specimens were selected by the orthogonal test [28,29]. Some representative points were selected from the comprehensive test. The test table is shown in Table 1.

#### 2.3.2. Specimens Grouping

In the design process, three influencing factors, including the thickness, the diameter of the holes, and the hole spacing, were considered. The test plan of the specimens is shown in Table 2 and Table 3. The compression tests include a one-way compression test and a two-way compression test.

#### 2.3.3. Test Procedure for Compressive Performance of Perforated Steel Plate Reinforced Concrete

Two-way compression

After curing for 28 days, the perforated steel plate reinforced concrete with a size of 150 mm × 150 mm × 150 mm was taken out from the curing space for the compressive test. In this paper, the two-way compression was to take the concrete face of the perforated steel plate reinforced concrete specimen as the bearing surface of the specimen. The two-way compression test is shown in Figure 4. Two concrete surfaces are used as the compression surface for the test. The loading speed of the testing machine is set to 0.3 MPa/s, and the load is continuously applied to the specimens [31].

2.One-way compression

In order to explore the relationship between the compressive strength of perforated steel plate reinforced concrete and the compression direction, this compression test also included the one-way compression of the specimen. In this paper, the one-way compression was to take the plate face of the perforated steel plate reinforced concrete specimen as the bearing surface of the specimen, as shown in Figure 5 below. The arrangement of the one-way compression is consistent with that of the two-way compression [31].

#### 2.3.4. Procedures for the Pull-Out Test of Perforated Steel Plate Reinforced Concrete

The pull-out tests were carried out on perforated steel plate reinforced concrete composite where the concrete was 150 mm × 150 mm × 150 mm and the perforated steel plate was 150 mm × 150 mm × 350 mm in size. The specimens reaching the standard age were taken out for the pull-out test. The pull-out test of the perforated steel plate reinforced concrete is shown in Figure 6 below. The loading speed of the testing machine to is set to 0.05 MPa/s, and the load is continuously applied to the specimens [31].

## 3. Test Results

### 3.1. Two-Way Compression Test of Perforated Steel Plate Reinforced Concrete

In the initial stage of the test, no cracks appeared on the surface of the perforated steel plate reinforced concrete through the outer perforated steel plate.

As the load continues to increase, the internal stress of the perforated steel plate reinforced concrete continues to accumulate, and cracks begin to appear on the surface of the specimens. The cracks first appear in the lower left corner and gradually expand through the entire surface of the specimens.

When the load is approaching the ultimate load, some debris falls off the surface, and the middle position of the perforated steel plate is separated from the concrete. Outward wrinkles are generated at the upper and lower 1/4 of the edge of the specimens. At this time, internal cracks can be observed through the outer perforated steel plate that develop through the concrete surface. The damage to the perforated steel plate reinforced concrete specimens is shown in Figure 7.

### 3.2. One-Way Compression Test of Perforated Steel Plate Reinforced Concrete

At the initial stage of the test, no change was observed on the specimen, and the load continued to increase. Cracks began to appear at the two corners above the concrete on the surface of the specimen and gradually developed until they penetrated the whole surface of the specimen. When the load is close to the ultimate load, the crack width of the concrete increases first; then, the steel plate with holes buckles, and the concrete breaks and falls. The damage process is shown in Figure 8 below.

### 3.3. Pull-Out Test of Perforated Steel Plate Reinforced Concrete

In the pull-out test, 27 groups of perforated steel–concrete specimens were tested. According to the damage to the specimens, the failure types of the specimens were classified into two categories.

The perforated steel plate was severely deformed, and the specimens were split and damaged.

This type of damage occurred when the thickness of the perforated steel plate was 0.5 mm. Because the perforated steel plate was too thin, the round holes on the steel plate were deformed under the tensile force. The round holes at the side of the plate were pulled off. With the increase in tensile force, the perforated steel plate was finally separated from the concrete, as shown in Figure 9 below.

2.The perforated steel plate retained its shape while the whole specimens were broken.

During the loading process, the perforated steel plate and the concrete gradually separated. At the end of the test, the perforated steel plate was completely pulled out of the concrete. The perforated steel plate had no obvious deformation. However, the concrete was broken into two parts under the tensile force, as shown in Figure 10.

### 3.4. Analysis of Test Results

#### 3.4.1. Two-Way Compression Test

3.Through calculation, it can be known that the data measured in the experiment were valid; therefore, these data can be used as comparisons for the finite element simulation.

The compressive strengths of all the specimens measured in this compression test are shown in Table 4.

When the parameters of the perforated steel plate reinforced concrete are selected according to test number 3, the volume ratio of the steel is 0.92%, and the compressive strength of the perforated steel plate reinforced concrete reaches 57.53 MPa, which is 97.0% higher than the 29.2 MPa of the plain concrete in the same batch. According to Ying’s [35] experimental research on steel fiber concrete, when the volume ratio of steel is 5%, the compressive strength of ultra-short and ultra-fine steel fiber concrete will reach the maximum value of 70.3 MPa, which is 65.8% higher than the 42.4 MPa of the same batch of plain concrete. The amount of steel used in the perforated steel plate reinforced concrete is 0.184 times that, and the improvement of the compressive strength is 31.2% greater than that. According to Lin’s [36] experimental research on steel fiber concrete, when the volume ratio of steel is 1.25%, the compressive strength of end hook-shaped steel fiber concrete will reach the maximum value of 48.9 MPa, which is 17.3% higher than the 41.7 MPa of the same batch of plain concrete. The amount of steel used in the perforated steel plate reinforced concrete is 0.736 times that, and the improvement in the compressive strength is 79.7% greater than that.

4.Curve fitting

The computer controlling the universal testing machine will automatically draw the load displacement curve in each test and process the data recorded by the computer. The abscissa is the vertical displacement of the cross beam of the press; the ordinate is the load applied by the press; the strain of the specimen is divided by the elevation of the test piece by the value of the abscissa; and the stress of the specimen is divided by the pressure area of the specimen by the value of the ordinate. The stress–strain curve of each group of specimens can be obtained by fitting the test data with Origin software, as shown in the following in Figure 11.

From the above in Figure 11a–c, it can be seen that when the stress is small, the strain increases with the increase in the stress, which is similar to the linear proportional increase. At this time, the specimen is in the stage of elastic deformation. With the continuous increase in stress, the curve shows a tendency towards an upward bulge, and the plastic deformation of the perforated steel plate reinforced concrete continues to increase. When the ultimate stress is reached, the stress does not increase and begins to decrease, and the strain continues to increase.

The Gauss formula is used to fit the stress–strain curve of the perforated steel plate reinforced concrete with the test data, as shown in Equation (1). It can be seen from the R^2^ of the fitting curves in Figure 11 above 0.95 that the curve fitting is consistent with the change trend of the test data.
(1)$${Y=y}_{0}+\left(\frac{A}{W*\mathrm{sqrt}\left(\frac{\pi }{2}\right)}\right)*\exp\left(-2*{\left(\frac{{x-x}_{c}}{W}\right)}^{2}\right)$$

In the equation, Y_0_ is the minimum y value on the curve; $\frac{W}{2}$ is the standard deviation; and $X_{c}$ is the mathematical expectation.

#### 3.4.2. Pull-Out Test

(1) Through calculation, it can be known that the data measured in the experiment were valid; therefore these data can be used as comparisons for the finite element simulation.

The ultimate pull-out bearing capacities of all the specimens measured in this pull-out test are listed in Table 5.

(2) The influence of the different factors on the pull-out ultimate bearing capacity of the perforated steel plate reinforced concrete specimen was studied by changing the thickness, aperture, and spacing of the perforated steel plate of the perforated steel plate reinforced concrete specimen.

It can be seen from Table 5 that when the hole diameter and spacing of the perforated steel plate are unchanged and when the thickness of the perforated steel plate is increased from 0.5 mm to 0.75 mm and 1.0 mm, the pull-out bearing capacity of the specimen is increased by 45.0% and 65.6%, respectively, indicating that increasing the thickness of the steel plate can significantly improve the pull-out bearing capacity of the specimen.

It can be seen from Table 5 that under the condition of keeping the plate thickness of the perforated steel plate unchanged, when the hole diameter and the hole spacing of the perforated steel plate are reduced from 25 mm and 15 mm to 20 mm and 10 mm, the pull-out bearing capacity of the specimen increases by 0.4%, indicating that the reduction of the hole diameter and the hole spacing has a weak impact on the pull-out ultimate bearing capacity of the specimen. When the hole diameter and hole spacing are increased from 25 mm and 15 mm to 15 mm and 5 mm, the pull-out bearing capacity of the specimen increases by 17.4%, This shows that the continuous reduction of the hole diameter and hole spacing will obviously improve the pull-out bearing capacity of the specimen.

## 4. Finite Element Simulation

### 4.1. Compression Test Model

(1)Material constitutive

In the finite element method, the stress–strain relationship of steel adopts the Mises yield criterion [37] and the William–Warnke five-parameter failure criterion [38] for concrete. The constitutive relationship diagram is shown in Figure 12 below.

(2)Parameter selection

Q235 steel and concrete with a strength grade of C30 were selected in this simulation. The material parameters of steel are shown in Table 6, and the material parameters of concrete are shown in Table 7.

(3)Selection of unit type

We performed several finite element simulations. Here, we present in detail the models where the plate thickness is 1 mm, the hole diameter is 20 mm, and the hole spacing is 10 mm. The perforated steel plate is modeled by Solid185, and the concrete is modeled by Solid65.

(4)Node connection, boundary conditions, and loading method

In the ANSYS finite element simulation produced by ANSYS Company in Pittsburgh, USA, the concrete and perforated steel plate are connected by the Merge command to make them become a whole to bear the force. This model simulates the concrete compression test; so, a full restraint is set at the center of the bottom, and a surface load is applied on the top.

(5)Meshing

The meshing is shown in Figure 13. In this way, the accuracy of the calculation results can be ensured, and meanwhile, the run time can be reduced as much as possible.

### 4.2. Simulation of Perforated Steel Plate Reinforced Concrete under Compression Force

The test results are compared with the simulation results of the ANSYS finite element model. The results verify the reliability of the ANSYS finite element model.

(1)Numerical fitting of test

The ANSYS finite element software is used to model and analyze the nine groups of test combinations selected by the orthogonal test method. See Figure 14 and Figure 15 for the strain and stress cloud diagram of the perforated steel plate reinforced concrete specimen.

(2)Strain analysis of perforated steel plate reinforced concrete specimens

A load value slightly larger than the test result is inputted into the ANSYS finite element model, and the maximum bearable load can be obtained by the recorded data in each load step. The strain cloud diagram of the perforated steel plate reinforced concrete (plate thickness of 1 mm, hole diameter of 20 mm, and hole spacing of 10 mm) is shown in Figure 16.

From Figure 16, the strain cloud diagram shows that the maximum deformation appears at the top, which is consistent with the phenomenon observed in the test. The maximum deformation of the perforated steel plate reinforced concrete is 2.88 mm in the simulation, but it is 3.26 mm in the test, which indicates some errors in the simulation. This is attributed to the fact that the actual test simulates the maximum deformation that the perforated steel plate reinforced concrete specimen can bear, while the actual stress state of the perforated steel plate reinforced concrete specimen in the composite structure is simulated in the finite element model. The constraint used in the finite element model is an ideal constraint state. By fixing the bottom of the perforated steel plate reinforced concrete specimen and applying the surface load on the top of the specimen, the concrete and the perforated steel plate bear the load together. The actual test is that the perforated steel plate reinforced concrete in the specimens will be separated from each other and deformed under the action of the pressure-testing machine.

(3)Stress analysis of perforated steel plate reinforced concrete specimens

According to the actual data, the perforated steel–concrete (plate thickness of 1 mm, hole diameter of 20 mm, and hole spacing of 10 mm) can withstand the maximum stress of 56.67 MPa. In order to obtain the stress value more accurately, a load value slightly larger than the actual test result is applied to the model, and the simulation results of each load step are recorded and outputted. In order to ensure the accuracy of the data, the appropriate number of iteration steps and the largest possible tolerance value are set. The maximum stress value that the model can bear is 53.16 MPa.

The stress cloud diagram of the perforated steel plate reinforced concrete specimens is shown in Figure 11. Compared with the actual test result in Figure 6, the stress is relatively concentrated at the upper and lower 1/3 of the outer edges of the perforated steel plate, which is the location where wrinkle failure occurs. It is consistent with the overall stress distribution of the model, indicating that the analysis result of the model is reliable.

(4)Error analysis

It can be seen from the above in Figure 17 that the deviation between the load on the model and the actual test results is no more than 10%. The main reason is that the ANSYS analysis cannot accurately simulate the concrete failure process during the actual test.

(5)Range analysis of influencing factors

The analysis steps are as follows: (i) Calculate Ki (i is the level number, taking 1, 2, 3). Ki is the sum of the different levels of test indicators corresponding to each factor. (ii) Sum up the 3 Ki corresponding to each factor to verify the accuracy of the calculation results. (iii) Calculate the average value of the ki corresponding to the Ki in each factor. (iv) Find the range value corresponding to the ki in each factor. (v) Determine the primary and secondary order of the factors according to the range value of each factor. (vi) Determine the optimal level of the factor according to the ki corresponding to each factor. (vii) Combine the optimal level of each factor and finally form the optimal test combination [39].

The range analysis of the compressive strength of the perforated steel plate reinforced concrete is shown in Table 8 (A (thickness of perforated steel plate); B (hole diameter of perforated steel plate); and C (hole spacing of perforated steel plate)). By comparing the range values corresponding to ki in the table, it can be obtained that in the compression test, the order of the three factors on the perforated steel plate reinforced concrete is A > B > C; that is, the factors having an impact on the compressive strength in an increasing order are hole spacing, the hole diameter, and the thickness of the steel plate. The range of the thickness of the perforated steel plate is 5.79, which is significantly larger than the range of the other factors. Therefore, the thickness has a significant influence on the compressive strength of the perforated steel plate reinforced concrete. By comparing the ki corresponding to each factor, it can be concluded that the optimal levels of each factor are A2, B1, and C1; that is, the optimal level is A2B1C1. The compressive strength of the perforated steel plate reinforced concrete reaches the maximum 56.8 MPa when the thickness of the perforated steel plate is 0.75 mm, the hole diameter of the perforated steel plate is 15 mm, and the hole spacing is 5 mm.

### 4.3. Establishment of Pull-Out Model

(1)Choose the appropriate unit type

The Solid185 element is selected in the model, and the Mises yield criterion is adopted as the failure criterion of the perforated steel plate [37]. The concrete element of the perforated steel plate reinforced concrete is Solid65, and the concrete follows the William–Warnke five-parameter failure criterion [38].

(2)Select material parameters

C30 concrete and Q235 steel are selected in the perforated steel plate reinforced concrete mode.

(3)Set constraints

Apply X-direction, Y-direction, and Z-direction displacement constraints to the model; this ensures that the displacement of the model in all directions is zero.

(4)Apply pull-out load

In the model, consider the actual stress condition of the perforated steel plate reinforced concrete specimens; the nodal load is applied to the perforated steel plate.

(5)Model of perforated steel plate reinforced concrete

The pull-out model includes two parts, i.e., the concrete model and the perforated steel plate model. The pull-out model of the perforated steel plate reinforced concrete is shown in Figure 18.

### 4.4. Simulation of Perforated Steel Plate Reinforced Concrete under Pull-Out Force

(1)Numerical fitting of test

ANSYS finite element software is used to model and analyze the nine groups of test combinations selected by the orthogonal test method. See Figure 19 and Figure 20 for the strain and stress cloud diagram of the perforated steel plate reinforced concrete specimen.

(2)Results analysis of pull-out model of perforated steel plate reinforced concrete specimens

In order to make the pull-out model more authentic, a pull-out load of 20 KN is applied, the appropriate load steps are set, and the calculation results of each load step are recorded. The ultimate pull-out capacity that the model can bear is 15.82 KN. The overall strain cloud diagram of the perforated steel plate reinforced concrete specimens and the strain cloud diagram of the concrete are shown in Figure 21a,b.

It can be seen from Figure 21a,b that the perforated steel plate produces vertical deformation on its contact surface under the action of the pull-out load. The deformation of the perforated steel plate on both sides is greater than that in the middle. The concrete at the junction of the perforated steel plate is partially swelled due to the pull-out effect and the bonding effect of the perforated steel plate. In the pull test, the deformation of the perforated steel plate on both sides is slightly larger than that in the middle. Under the action of tensile stress, the concrete will break and produce small deformation, while the other parts of the concrete will not deform and will remain intact, which is consistent with the deformation in the model.

(3)Range analysis of influencing factors

The range analysis of the ultimate tensile bearing capacity of the perforated steel plate reinforced concrete is shown in Table 9 [40,41,42]. By comparing the range values corresponding to ki in the table, it can be obtained that in the perforated steel plate reinforced concrete pull-out test, the order of the influence of each factor is A > B > C, which means the impact factors from small to large are the spacing between the holes of the perforated steel plate, the hole diameter of the perforated steel plate, and the thickness. Because the range value between factors A and B is quite different from that of factor C, the hole spacing has a less significant influence on the ultimate bearing capacity of the perforated steel plate reinforced concrete. By comparing the ki corresponding to each factor, it can be concluded that the optimal levels of each factor are A3, B1, and C3; that is, the optimal combination is A3B1C3, indicating that the ultimate pull-out bearing capacity of the perforated steel plate reinforced concrete reaches the maximum value of 15.82 KN when the thickness of the perforated steel plate is 1.0 mm, the hole diameter is 15 mm, and the hole spacing is 15 mm.

## 5. Conclusions

This paper mainly studies the mechanical properties of perforated steel plate reinforced concrete. The compression test and pull-out test of the perforated steel plate reinforced concrete specimen were carried out, and the mechanical characteristics and failure characteristics of the specimen under a load were studied. The mechanical properties of the perforated steel plate reinforced concrete specimen and the effects of the steel thickness, pore diameter, and pore spacing on the mechanical properties of the perforated steel plate reinforced concrete specimen were studied by numerical simulation using ANSYS software.

The main conclusions of this article are as follows:(1)In the compression test, the failure characteristics of the perforated steel–concrete specimens are quite different from those of the ordinary concrete specimens. When the ultimate load is reached, the perforated steel–concrete specimen basically retains its original shape. The compressive strength of the perforated steel plate reinforced concrete specimen is twice that of the ordinary concrete specimen. The total amount of steel used for the multi cavity steel plate–concrete composite floor slab is nearly twice that of the existing prefabricated floor slab, and the corresponding bearing capacity is also increased by more than four times [33].(2)The models established by the ANSYS finite element analysis are consistent with the test data, which confirms that the modeling analysis is reliable and can be used for further research. The significance of the influence of various factors on the compressive strength and ultimate tensile strength of the perforated steel plate reinforced concrete is the thickness of the perforated steel plate, the hole diameter, and the hole spacing, in decreasing order.(3)The optimal level of the compressive strength of the perforated steel plate reinforced concrete is A2B1C1; that is, the thickness of the perforated steel plate is 0.75 mm, the hole diameter is 15 mm, and the hole spacing is 5 mm.(4)The optimal level of the tensile ultimate bearing capacity of the perforated steel plate reinforced concrete is A3B1C3; that is, the thickness of the perforated steel plate is 1.0 mm, the hole diameter is 15 mm, and the hole spacing is 15 mm.

## Figures and Tables

**Figure 1 materials-15-06944-f001:**
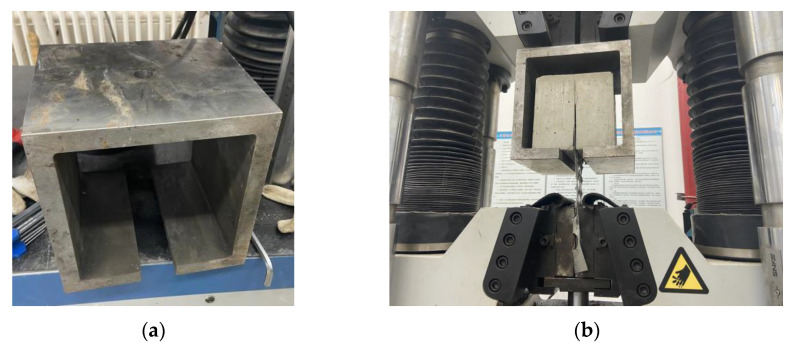
Pull-out test of perforated steel plate reinforced concrete: (**a**) drawing fixture; (**b**) pull-out test.

**Figure 2 materials-15-06944-f002:**
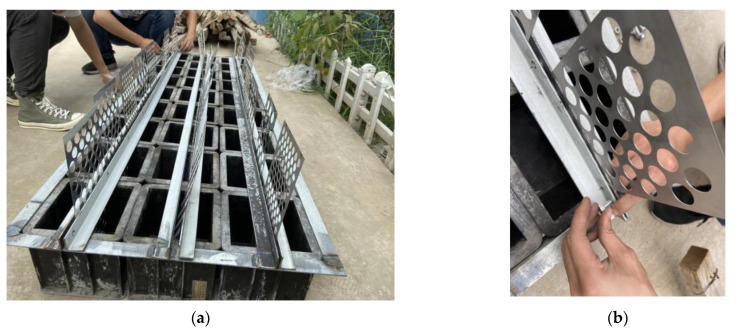
Test mold: (**a**) fix of test mold; (**b**) details of test mold.

**Figure 3 materials-15-06944-f003:**
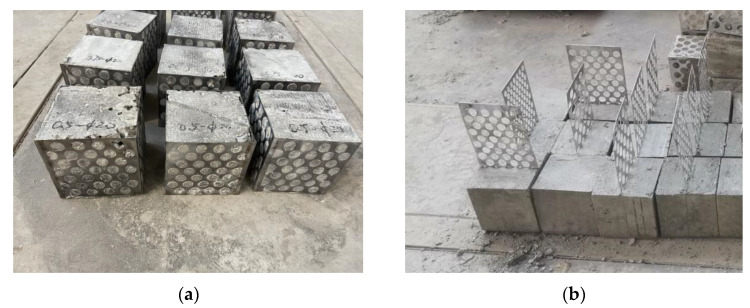
Specimens: (**a**) compression specimens; (**b**) pull-out specimens.

**Figure 4 materials-15-06944-f004:**
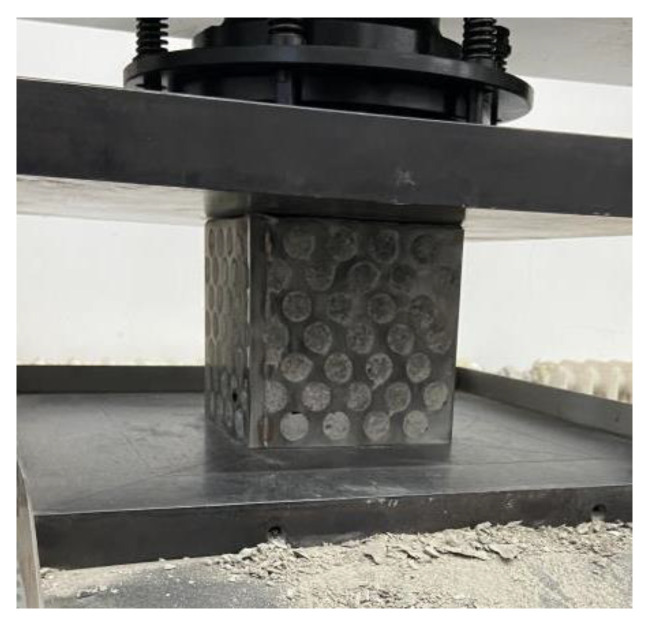
Two-way compression test.

**Figure 5 materials-15-06944-f005:**
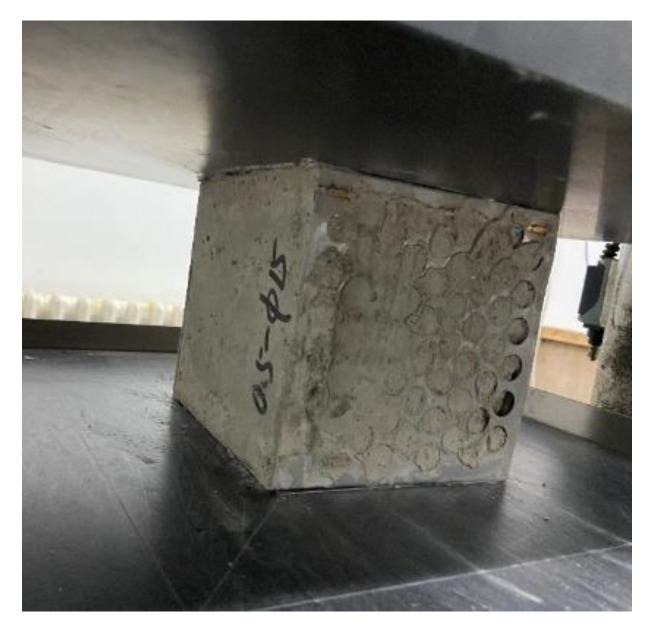
One-way compression test.

**Figure 6 materials-15-06944-f006:**
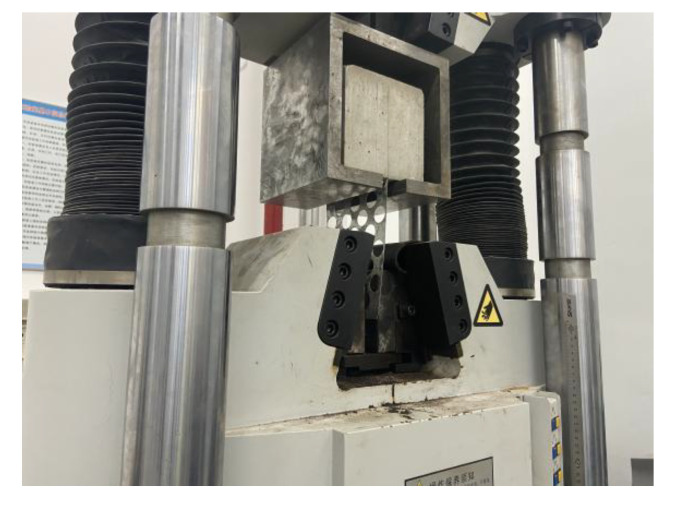
Pull-out test on perforated steel plate reinforced concrete.

**Figure 7 materials-15-06944-f007:**
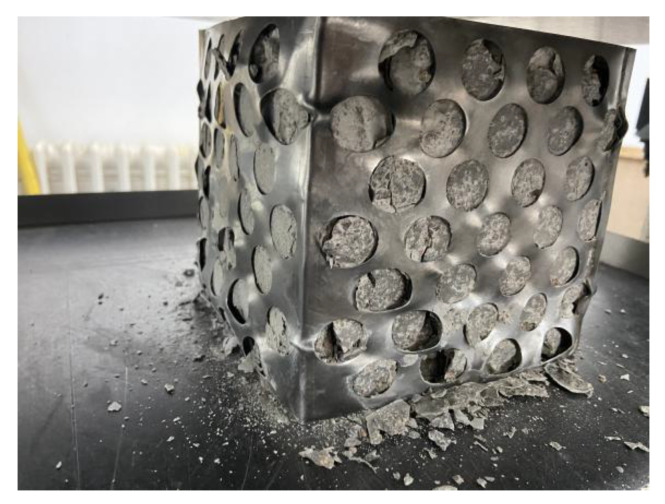
Block damage.

**Figure 8 materials-15-06944-f008:**
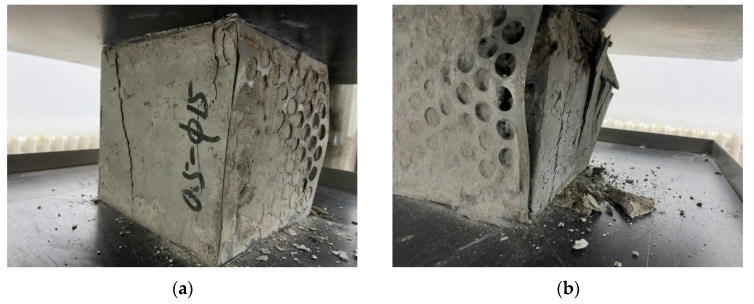
Block damage: (**a**) the surface of the specimen is penetrated by cracks; (**b**) the specimen is damaged.

**Figure 9 materials-15-06944-f009:**
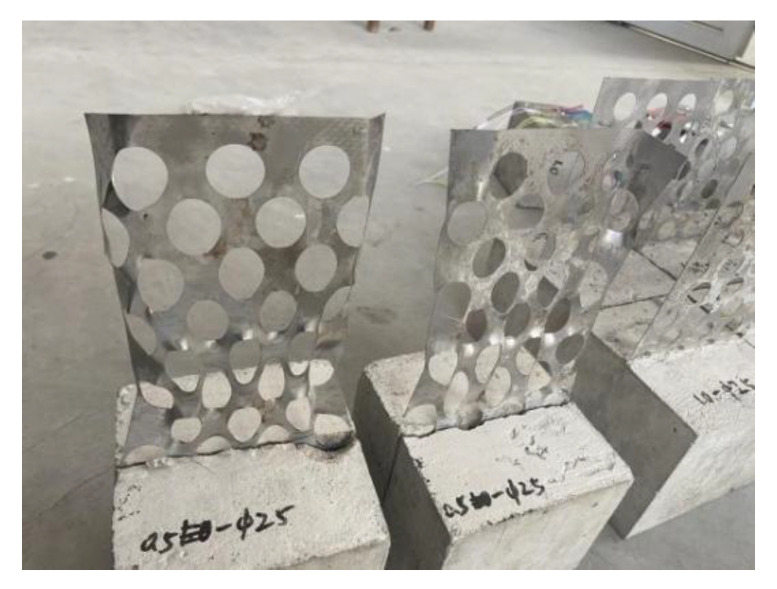
The first kind of destruction.

**Figure 10 materials-15-06944-f010:**
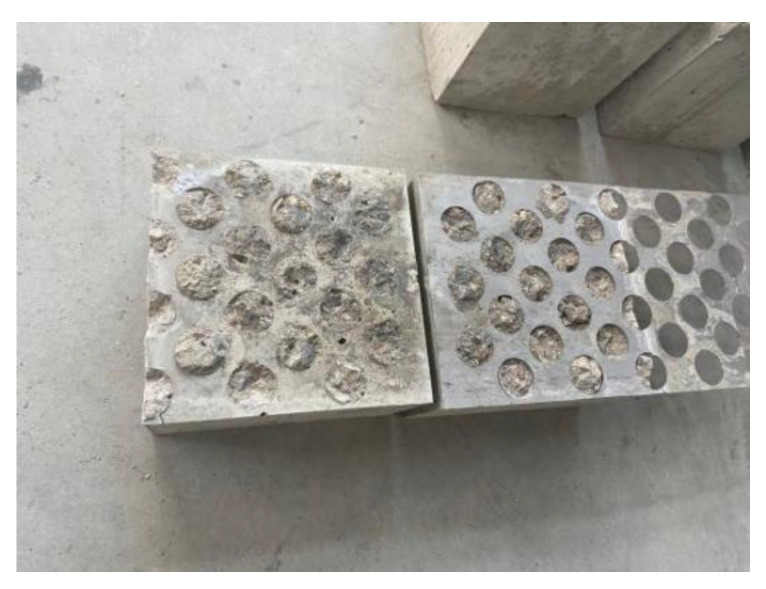
The second kind of destruction.

**Figure 11 materials-15-06944-f011:**
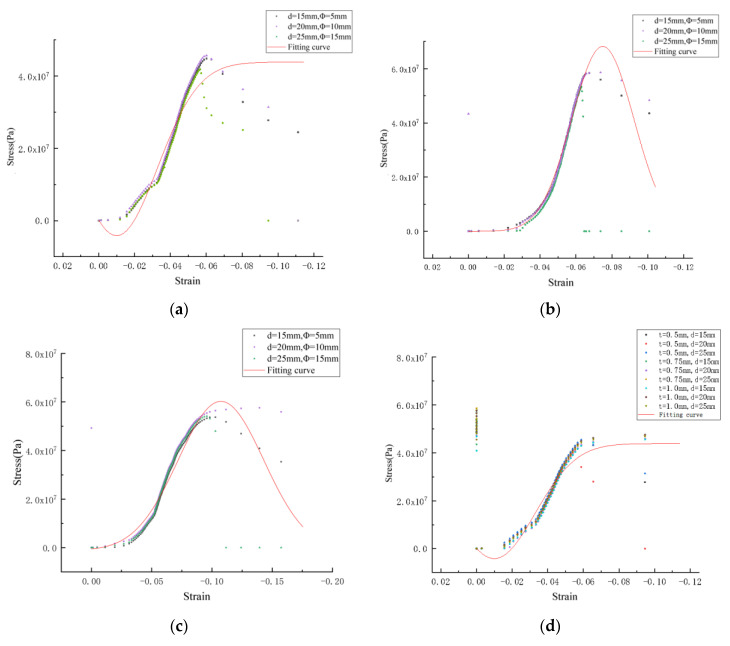
Stress–strain curve of perforated steel sheet concrete: (**a**) thickness of 0.5 mm; (**b**) thickness of 0.5 mm; (**c**) thickness of 0.5 mm; (**d**) summary of three thicknesses.

**Figure 12 materials-15-06944-f012:**
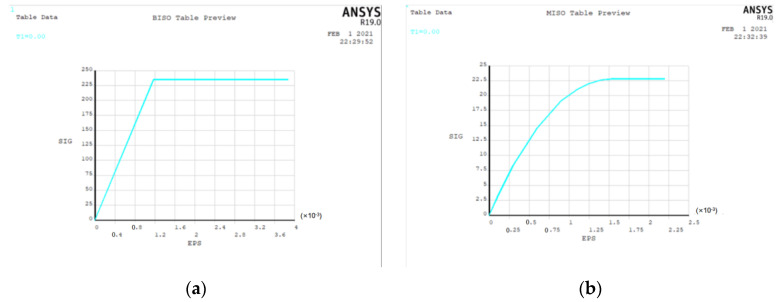
Constitutive relation of materials: (**a**) steel; (**b**) concrete.

**Figure 13 materials-15-06944-f013:**
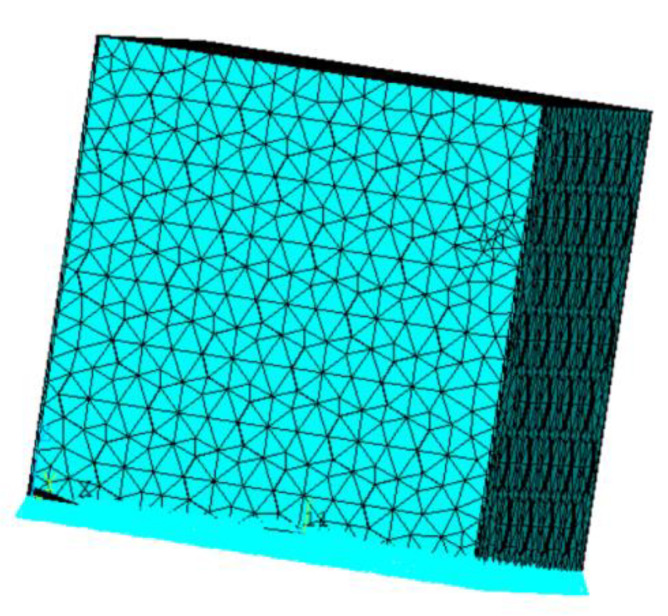
Meshing of perforated steel plate reinforced concrete.

**Figure 14 materials-15-06944-f014:**
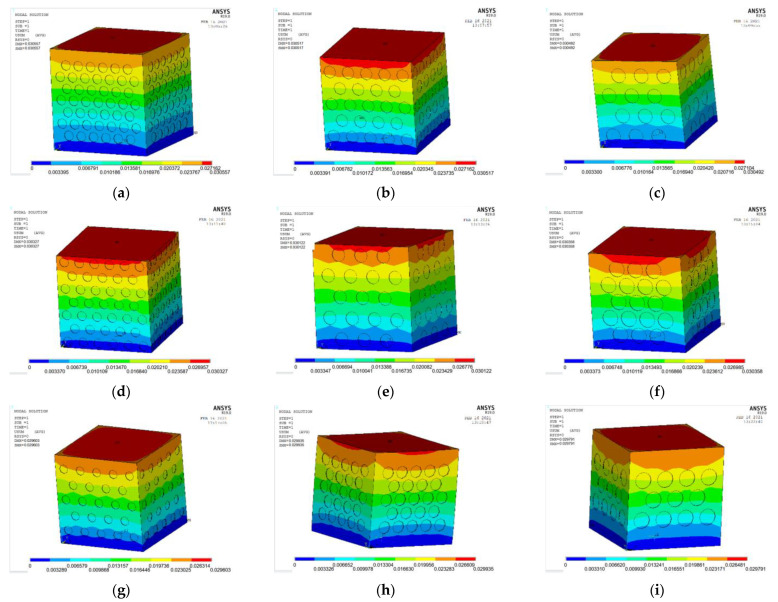
Strain cloud images of nine groups of compression perforated steel plate reinforced concrete samples: (**a**) test number 1; (**b**) test number 4; (**c**) test number 7; (**d**) test number 2; (**e**) test number 5; (**f**) test number 8; (**g**) test number 3; (**h**) test number 6; (**i**) test number 9.

**Figure 15 materials-15-06944-f015:**
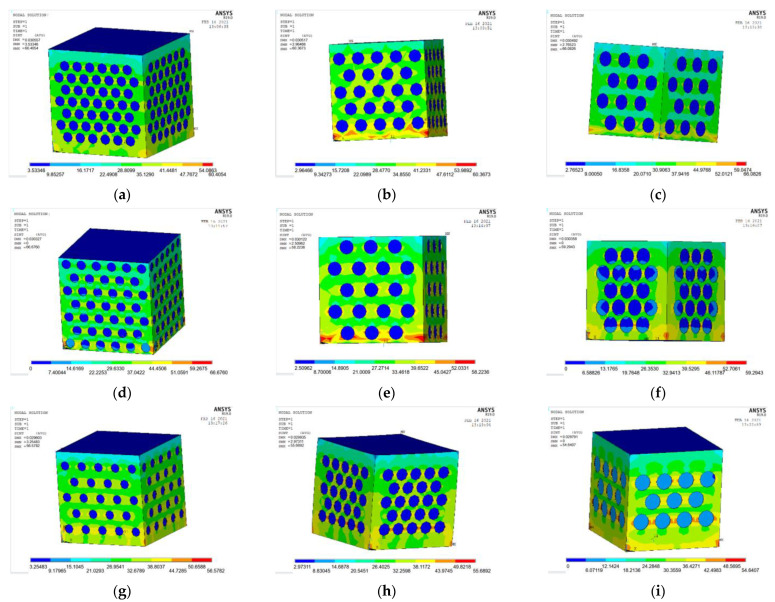
Stress cloud images of nine groups of compression perforated steel plate reinforced concrete sam-ples: (**a**) test number 1; (**b**) test number 4; (**c**) test number 7; (**d**) test number 2; (**e**) test number 5; (**f**) test number 8; (**g**) test number 3; (**h**) test number 6; (**i**) test number 9.

**Figure 16 materials-15-06944-f016:**
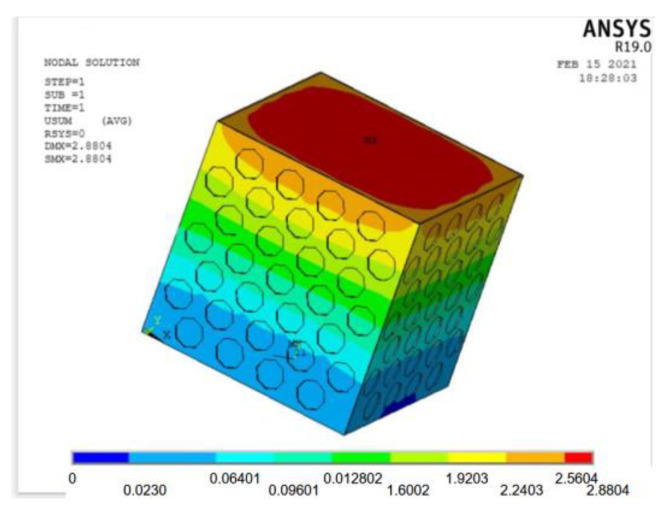
Strain nebula of perforated steel plate reinforced concrete model.

**Figure 17 materials-15-06944-f017:**
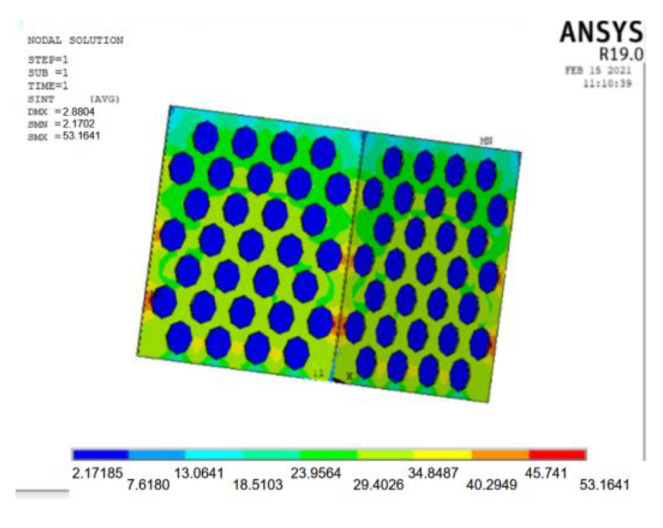
Stress diagram of perforated steel plate reinforced concrete model.

**Figure 18 materials-15-06944-f018:**
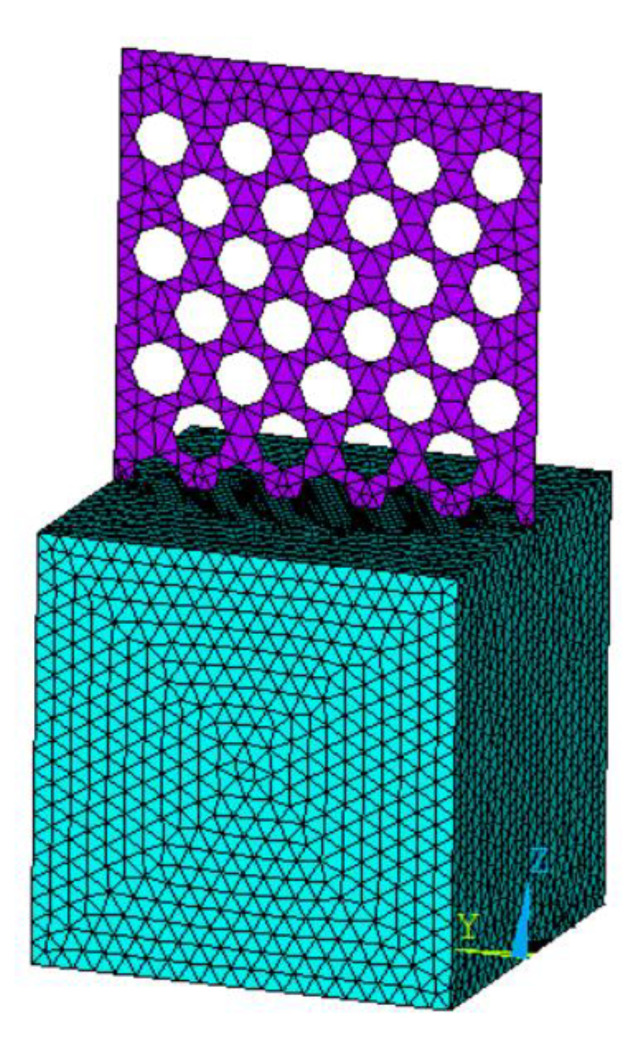
Pull-out model of perforated steel plate reinforced concrete.

**Figure 19 materials-15-06944-f019:**
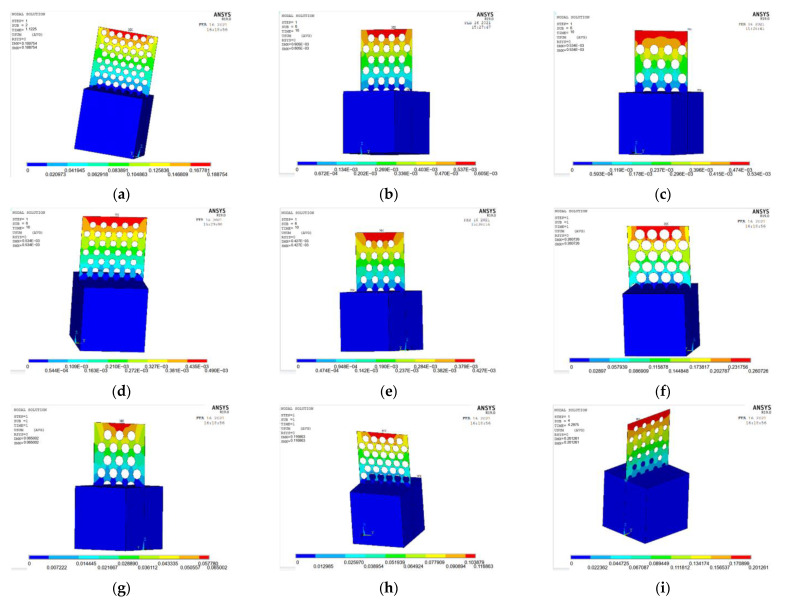
Strain cloud images of nine groups of perforated steel plate reinforced concrete samples: (**a**) test number 1; (**b**) test number 4; (**c**) test number 7; (**d**) test number 2; (**e**) test number 5; (**f**) test number 8; (**g**) test number 3; (**h**) test number 6; (**i**) test number 9.

**Figure 20 materials-15-06944-f020:**
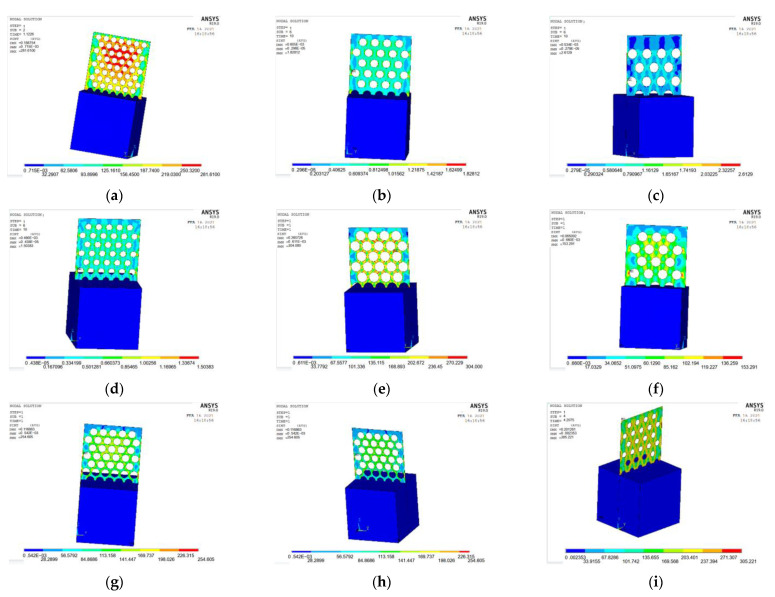
Stress cloud images of nine groups of perforated steel plate reinforced concrete samples: (**a**) test number 1; (**b**) test number 4; (**c**) test number 7; (**d**) test number 2; (**e**) test number 5; (**f**) test number 8; (**g**) test number 3; (**h**) test number 6; (**i**) test number 9.

**Figure 21 materials-15-06944-f021:**
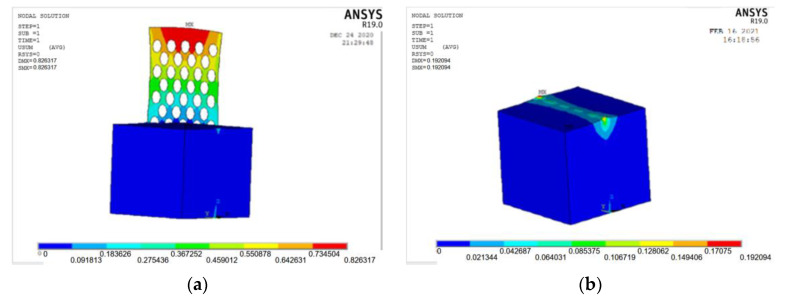
Cloud diagram: (**a**) strain cloud diagram of specimens; (**b**) concrete strain cloud map.

**Table 1 materials-15-06944-t001:** Test table.

Test Number	Thickness of Plate (mm)	Hole Diameter (mm)	Hole Spacing (mm)
1	0.5	15	5
2	0.5	20	10
3	0.5	25	15
4	0.75	15	5
5	0.75	20	10
6	0.75	25	15
7	1.0	15	5
8	1.0	20	10
9	1.0	25	15

**Table 2 materials-15-06944-t002:** Compression test program table.

Test Number	Specimen Size (mm)	Curing Age (mm)	Specimen Quantity (Pieces)	Specimen Usage
1	150 × 150 × 150	28	6	Compression test
2				
3				
4	150 × 150 × 150	28	6	Compression test
5				
6				
7	150 × 150 × 150	28	6	Compression test
8				
9				

**Table 3 materials-15-06944-t003:** Summary of pull-out test scheme.

Test Number	Specimen Size (mm)	Curing Age (Days)	Specimen Quantity (Pieces)	Specimen Usage
1	150 × 150 × 150	28	3	Pull-out test
2				
3				
4	150 × 150 × 150	28	3	Pull-out test
5				
6				
7	150 × 150 × 150	28	3	Pull-out test
8				
9				

**Table 4 materials-15-06944-t004:** Compressive strengths of two-way compression specimens.

Test Number	Thickness of Plate (mm)	Hole Diameter (mm)	Hole Spacing (mm)	Mean of Compressive Strength (MPa)	Standard Deviation
1	0.5	15	5	53.60	2.30
2	0.5	20	10	44.22	1.63
3	0.5	25	15	54.58	4.18
4	0.75	15	5	56.84	2.43
5	0.75	20	10	56.16	3.76
6	0.75	25	15	53.15	5.79
7	1	15	5	55.20	1.68
8	1	20	10	55.01	3.84
9	1	25	15	50.74	9.17

**Table 5 materials-15-06944-t005:** Summary of ultimate pull-out bearing capacity of specimens.

Test Number	Thickness of Plate (mm)	Hole Diameter (mm)	Hole Spacing (mm)	Mean of Ultimate Load (kN)	Standard Deviation
1	0.5	15	5	9.53	0.27
2	0.5	20	10	8.16	0.27
3	0.5	25	15	14.33	2.01
4	0.75	15	5	13.83	0.75
5	0.75	20	10	11.83	0.61
6	0.75	25	15	11.78	0.23
7	1	15	5	14.33	2.01
8	1	20	10	13.73	2.35
9	1	25	15	13.48	2.13

**Table 6 materials-15-06944-t006:** Steel material parameters.

Steel Model	Yield Strength (MPa)	Density (g/cm^3^)	Modulus of Elasticity (MPa)	Poisson’s Ratio
Q235	235	7.85	$2.1\times 10^{5}$	0.33

**Table 7 materials-15-06944-t007:** Concrete material parameters.

Concrete Model	Compressive Strength (MPa)	Density (kg/mm^3^)	Modulus of Elasticity (MPa)	Poisson’s Ratio
C30	30	$2.50\times 10^{-6}$	$3\times 10^{4}$	0.2

**Table 8 materials-15-06944-t008:** Range analysis results of compression test of perforated steel sheet concrete.

Range Calculation	Test Factors		
	A	B	C
K1	160.81	173.02	170.75
K2	178.18	164.33	165.74
K3	166.89	168.53	169.39
Sum	505.88	505.88	505.88
k1	53.60	57.67	56.91
k2	59.39	54.77	55.24
K3	55.63	56.17	56.46
Range R	5.79	2.897	1.67
Primary and secondary factors	A > B > C		
Optimal level	A2	B1	C1
Optimal combination	A2B1C1		

**Table 9 materials-15-06944-t009:** Range analysis results of pull-out test of perforated steel sheet concrete.

Range Calculation	Test Factors		
	A	B	C
K1	26.84	40.79	34.28
K2	37.41	32.57	35.02
K3	40.7	31.59	35.65
Sum	104.95	104.95	104.95
k1	8.94	13.59	11.42
k2	12.47	10.85	11.67
K3	13.56	10.53	11.88
Range R	4.62	3.06	0.45
Primary and secondary factors	A > B > C		
Optimal level	A3	B1	C3
Optimal combination	A3B1C3		

## Data Availability

The raw/processed data required to reproduce these findings cannot be shared at this time as the data also forms part of an ongoing study.

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
