# Peer review of "Study on the Mechanical Properties of Perforated Steel Plate Reinforced Concrete"

_materials, 2022, doi:10.3390/ma15196944_

Round 1

Reviewer 1 Report

This manuscript presented experimentally and numerically the mechanical properties of porous steel plate-concrete. The manuscript needs thorough corrections before acceptance. The following are specific comments:
1. There are many sentences in the text that have errors in grammar and should be corrected. The authors suggest doing a proof of English reading and editing a manuscript to correct all grammar errors.
2. The abstract has a poor presentation, it should be re-written to include the studied parameters, the methodology, and the important findings.
3. On what basis do the authors consider the range or level of each of the experimental variables? Would you please elaborate on the manuscript to better understand the readers?
4. The materials' mechanical properties (Steel plates and concrete) should be provided with a reasonable explanation (for example, the stress-strain curves for each property measured, such as compressive strength of concrete, the tensile strength, etc.).
5. You should write all standards used in testing different specimens.
6. The constitutive model used in FEM should be explained in detail.
7. More recent references should be added to the introduction such as :
1. Development of shear capacity equations for RC beams strengthened with UHPFRC.
2.Numerical analysis of the shear behavior of FRP-strengthened continuous RC beams having web openings.
3.Behavior of RC beams strengthened in shear with ultra-high performance fiber reinforced concrete (UHPFRC).
4.Behavior of steel I-beam embedded in normal and steel fiber reinforced concrete incorporating demountable bolted connectors.
5. Bond behavior between concrete and prefabricated Ultra High-Performance Fiber-Reinforced Concrete (UHPFRC) plates.
6.Effect of interfacial surface preparation technique on bond characteristics of both NSC-UHPFRC and NSC-NSC composites.
7.Flexural strengthening of RC one way solid slab with Strain Hardening Cementitious Composites (SHCC).
8.What the effect of the mesh size on the model effeciency? the authors must compare their findings with previous studies such as (Numerical analysis of the shear behavior of FRP-strengthened continuous RC beams having web openings).
9. How can the results of this paper help to improve the specification?

Reviewer 2 Report

The study presents a new composite construction material. As seen in the introduction section, this is the first publication in relation to this new material, where concrete is reinforced with porous steel-plates.

The comments made by this author are as follows;

When citing references by the name of the author, no present in capital letter the first names of the authors, just the surname.

In line 68, the number of the reference should be next to the author.

No one of the results were compared with the ones presented in the introduction section. At least, a comparison regarding the reinforcement percentage could be done with the ones exposed in the introduction section or/and with conventional ones.

In figure 1 (a) font size should be checked.

Did the situation in lines 102-104 occur?

Figure 3, shows the location of the mesh for compression test. It comes to my mind many questions regarding why you chose that mesh disposition, but the main one is; it looks like there is no spacer block, is that possible? Steel reinforcement should be embedded by an specific distance, depending on the conditions.

In table 2 it says “Specimens quantity”, Three replicates are made for each specimen is mentioned before, so, what does it mean?

In table 2 Specimen number seems to be a bit confusing. In table 1, the designation of each specimen is made, so used that one.

In line 128, it is said biaxial compression, but there is load in one axis. That should be clarify.

In figure 6, it is seen the specimen after compression test with the concrete block and the porous steel-plates attached, but in in line 130, when describing the compression test, it is said that the porous steel plate were taken out. This is an incoherence.

The results in section 3.2 and 3.4 are not discussed and no benchmarking was done whatsoever.

The mean and the deviation standard should be presented in table 4 and 5 instead of all the results.

In line 168 is said “This type of damage occurred when the thickness of the porous steel plate was less than 0.5mm”, but the minimum thickness is 0.5 mm, this incoherence should be clarify.

Figure 7 and 8 shows a type of failure after pull-out test. These should be relocated after results. It should be interesting (almost mandatory) to know what type of failure took place foe each specimen.

In Figure 10, it can be seen the results of deformation in the model for compression. There is incoherence since the load is applied by steel rigid plates but on the top of the block, there are different deformation, greater in the center and lower on the edge, as far as this reviewer knows, that is not possible, same deformation takes place for centre and edge on the top (and on the bottom) of the sample.

In line 219 is mentioned the deformation in the compression test. This result is not presented in the corresponding section. So, deformation should be added for each of the sample.

Furthermore, how deformation was measured, as seen in the figure X, it seems that it comes from the crossbar of the press, am i right?

Round 2

Reviewer 1 Report

The manuscript has been correctly modified except the author list of reference No.  26. Only four authors are the author list (Sakr, M.A.; Sleemah, A.A.; Khalifa, T.M.; Mansour, W.N.;).

Reviewer 2 Report

Thanks for considered the comments made by this reviewer. The quality of the paper has been improved.

Regarding the location of the mesh for the compressive strength test, it seems that mesh is the main role for supporting the load. I must insist that the placing of the mesh is not adequate for determining the performance in the compressive strength test. The mesh should have been embedded.

This reviewer did not succeed in understanding the one-way and the two-ways compressive strength tests, and the response 10 “… were taken out for the compressive test, not separate porous steel plate.” Could you explain in details both of them? Sorry for the inconvenience.

Regarding the response 11, data has been analysed but, is not any other research to compare with? Are not any reference to compare the results to? This is what I meant when I used the term benchmarking

For further submission, this reviewer suggests that the response to the reviewer should be given in details and an easier way to follow the changes made by the authors.
